# Polysiloxane Hybrids via Sol-Gel Process: Effect of Temperature on Network Formation

**Maria Criado** [1,2,*] ，**Isabel Sobrados** [2] **and Jesus Sanz** [2]

[1]  Instituto de Ciencias de la Construcción Eduardo Torroja, CSIC, Serrano Galvache 4, 28033 Madrid, Spain
[2]  Instituto de Ciencia de Materiales de Madrid, CSIC, Sor Juana Inés de la Cruz 3,
    28049 Cantoblanco-Madrid, Spain; isobrado@icmm.csic.es (I.S.); jsanz@icmm.csic.es (J.S.)
*  Correspondence: maria.criado@ietcc.csic.es; Tel.: +34-91-302-04-40; Fax: +34-91-302-07-00

**Abstract:** The effect of temperature on the network formation of polysiloxane hybrids was evaluated since this type of material is currently in high demand. In the last decades, the deposition of these coatings on different substrates, mostly metals, has demonstrated anticorrosion properties. Sol-gel hybrids were prepared by mixing 3-methacryloxypropyltrimethoxysilane (MPTS) and tetramethyl orthosilicate (TMOS) with a molar ratio of 1. The formation, thickness and composition of these hybrid materials were evaluated by nuclear magnetic resonance (NMR) spectroscopy and scanning electron microscopy (SEM), respectively. The results showed that the temperature plays an important role in the network formation, the total condensation degree and the total dimensionality of the hybrid materials. At room temperature, the polysiloxane hybrids presented total condensation degrees lower than 75% and a total dimensionality ($d_{total}$) = 2.5, while those obtained at 65 °C presented total condensation degrees higher than 80% and a $d_{total}$ = 2.8. The ideal conditions to prepare polysiloxane hybrids are 65 °C for 4 h, where this shows a higher atomic percentage of Si and a greater thickness.

**Keywords:** organic-inorganic hybrids; sol-gel synthesis; reaction temperature; NMR; polymerization; total dimensionality

## 1. Introduction

The sol-gel process in the liquid phase consists of the creation of an oxide network by condensation reactions of molecular precursors [1–4]. The inorganic and organic methods are the two main ones to prepare sol-gel coatings. In the inorganic method a network is formed in the continuous liquid medium through the formation of a colloidal suspension (usually oxides) and gelation of the sol (colloidal suspension of particles with size from 1 to 100 nm), while the organic method consists in a solution of monomeric metal or metalloid alkoxide precursors $M(OR)_n$; $R$ is typically an alkyl group ($C_xH_{2x+1}$) and M represents a network- forming element, such as Si, Ti, Zr, Al, Fe, B, etc., in an alcohol or other low-molecular weight organic solvent [2,5–8].

Synthesis of the organic-inorganic materials by the sol-gel process offers some advantages such as mild processing conditions, high purity, homogeneity of products, low temperatures and the possibility of modifying the process conditions. In these hybrid materials, the organic component is incorporated in an inorganic network to achieve a product with unique morphology, size and multifunctional properties associated with the interaction between both components [9,10].

The sol-gel process is fundamentally affected by the initial reaction conditions, such as pH of the medium, reaction temperature, solvent composition and the molar ratio of reactants among others [11–15]. Sarmento, et al. [11,16] carried out several studies to know the effect of siloxane content, pH of initial sol and catalyst content on the structure of the dried gels trough small-angle X-ray scattering (SAXS) in the synthesis of siloxane-polymethylmethacrylate (PMMA) hybrids prepared via

sol-gel. Isolated clusters that are spatially correlated are formed by siloxane groups located at the ends of PMMA chains. They observed that a greater amount of 3-methacryloxypropyltrimethoxysilane (MPTS)/tetramethyl orthosilicate (TMOS) leads to a reduction of the average intercluster distance owing to the increasing number of siloxane groups. A more efficient polymerization of methylmethacrylate (MMA) monomers but no noticeable effect on the average intercluster distance was promoted by the increase of catalyst content, benzoyl peroxide (BPO). In addition, polycondensation reactions between silicon species of both TMOS and MPTS silicon alkoxides are favored at high pH, leading to a structure in which PMMA chains were bonded to all siloxane clusters.

In previous works, the authors also evaluated the effect of the nature of reagents and the molar ratios of the polysiloxane hybrid films on the corrosion process in carbonated simulated concrete pore solution or contaminated with chloride [12,17]. The results indicated that all the coatings improved the corrosion resistance of carbon steel in an alkaline environment contaminated with 3 wt.% NaCl solution and showed that the protective properties of the coating were especially improved when methyltriethoxysilane (MTES) was added to the formulations and when the TMOS/MPTS mixture was used; in a carbonated synthetic solution, the TMOS/MTES mixture presented the best protective properties. Highly cross-linked network and greater adhesion to the metallic surface of the mixtures containing MTES restrict corrosion attack. In addition, this attack is delayed with a higher molar ratio. The nature of the reagents used in the synthesis of the coatings had a greater influence than the molar ratio on the anticorrosion features of these films.

Bakhshandeh, et al. [13] evaluated the effect of organic solvents on the formation of silica domains in a series of epoxy-silica hybrid composites. Taking into account appearance, thermomechanical characteristics and the dispersion of silica domains throughout the epoxy, a mixture of xylene and ethanol (3:1) was properly comparable with tetrahydrofuran (THF) solvent. An increase of the solvent content implied enhanced thermal stability.

The reaction temperature is considered essential for corrosion protection of metals; during heat treatment, condensation of silanol groups results in the formation of Si–O–Si siloxane chains. Crosslinking and branching of these groups give rise to the formation of a dense network, which enhanced barrier properties due to the limitation of electrolyte access to the underlying metal or alloy [18]. Similar conclusions were reported by Kunst, et al. [19] where thermal curing at 90 °C for 20 min was sufficient to lead to the formation of a film with a regular, homogenous and compact structure and great flexibility. This film was obtained from a sol that consisted of two alkoxide precursors, MPTS and TEOS, with nitrate cerium and polyethylene glycol plasticizers (20 g·L$^{-1}$) and was subsequently deposited over galvanized steel, providing the best anticorrosive properties.

The temperature exerts a great effect on the reaction rate and kinetics for both hydrolysis and condensation reactions, and in extension in the network formation of polysiloxane hybrids, final sol-gel product and its final properties. In previous works [12], the authors observed that an MPTS and TMOS mixture with a molar ratio of 1 heated at 65 °C for 24 h and cured at 160 °C for 3 h allowed to obtain a protective coating against chloride ions. Modifying the temperature and reaction time, it was possible to control the polymerization degree and the thickness of the coatings with excellent final properties in an energy-efficient and economically viable way. In this work, coatings were prepared at 65 °C with short reaction times and at room temperature using long reaction times, thus allowing the formation of polysiloxane hybrids without any thermal curing.

A suitable technique for the study of the evolution of the network of polysiloxane hybrids over time at different reaction temperatures could be nuclear magnetic resonance spectroscopy (NMR). This spectroscopy has been a useful tool to study the polycondensation as well as the radical type of the polyaddition mechanism [20]. NMR spectroscopy was also used to estimate the scale of the miscibility of organic-inorganic hybrids through relaxation time measurements, which are sensitive to short-range interactions [21]. Besides, the influence of hydrolysis water ratio and organic content in the structure of sol-gel derived hybrid materials [22], the changes in chemical bonding and structure during the synthesis of organic-inorganic hybrid sol-gels [23,24], the effect of the siloxane content,

adjusted by addition of tetraethyl orthosilicate (TEOS), on the structure and thermal stability of the dried gels [25], the control of OH group concentration in the sol formation of a hybrid system [26], were followed using NMR spectroscopy.

Therefore, in this study, better knowledge of the effect of temperature (65 °C and room temperature) on the network formation of polysiloxane hybrids synthesized with 3-methacryloxypropyltrimethoxysilane (MPTS) and tetramethyl orthosilicate (TMOS) using a molar ratio of 1 was evaluated through NMR spectroscopy, specifically liquid and solid-state $^{29}$Si and $^{13}$C NMR have been applied to study the hydrolysis and the condensation mechanisms in these protective coatings.

## 2. Experimental

3-methacryloxypropyltrimethoxysilane ($CH_2$=C($CH_3$)COO($CH_2$)$_3$Si(OCH$_3$)$_3$, MPTS, 98%, Sigma-Aldrich, St. Louis, MO, USA) and tetramethyl orthosilicate (Si(OCH$_3$)$_4$, TMOS, 98%, Sigma-Aldrich) were employed to synthesize the polysiloxane hybrid materials via the sol-gel process. For that, a [MPTS]/[TMOS] molar ratio of 1, a $HNO_3$-acidified water ($HNO_3$, 70%, Sigma Aldrich, pH = 1 and [$H_2O$]/[Si] = 3.5), ethanol (EtOH, absolute, Sigma-Aldrich, [ethanol]/[$H_2O$] = 1) and a thermal initiator of polymerization (BPO, with 25% of water for synthesis, VWR International Eurolab, [BPO]/[MPTS] = 0.01) were employed. The presence of BPO is necessary since it can produce active centers for polymerization, decreasing the inhibition and retardation effects. The amounts of the reagents, the catalyst, the solvent, the molar ratio and the initial pH value were the same as those used in previous studies [12,17,27].

Two methodologies were employed to study the effect of the temperature on the hydrolysis and condensation mechanisms. In the first one, the polysiloxane hybrids were directly analyzed by nuclear magnetic resonance (NMR, Karlsruhe, Germany) spectroscopy at different times (up to 60 days), while on the other hand, these materials were also dried at 65 °C for 4, 24, 72 h, 7 and 11 days and analyzed by NMR spectroscopy with the same purpose.

Liquid-state $^{13}$C and $^{29}$Si NMR spectra and solid-state $^{13}$C CPMAS- and $^{29}$Si MAS-NMR spectra were recorded using a Bruker Avance-400 pulse spectrometer (Karlsruhe, Germany). Spectra were recorded after irradiation of samples with a π/2 (5-μs) pulse. The resonance frequencies used were 100.63 and 79.5 MHz, (9.4 T magnetic field). In order to avoid saturation effects, the recycle delay times used was 10 seconds. The spinning rate used in MAS-NMR experiments was 10 kHz. A contact time of 2 milliseconds and a recycle delay of 5 s were used in $^{13}$C CPMAS-NMR experiments. All measurements were taken at room temperature with TMS (tetramethylsilane) as an external standard. The error in chemical shift values was estimated to be lower than 0.5 ppm. NMR spectra deconvolutions were performed by using the DMFIT software (version 2015) [28]. Chemical shift (position of the line), intensity, width at half height and line shape (Lorentzian or Gaussian) of the components were deduced.

The spin-lattice relaxation times of silicate and aluminosilicates species in solution or solid compounds may vary widely and can be extremely long. Therefore, care should be exercised if quantitative data are to be derived from the peak intensities, and variable pulse repetitions or pulse width experiments are indispensable if reliable information on T1 is not available.

The experiments with increasing repetition time are difficult to evaluate in this work due to the few accumulations that can be registered between two consecutive kinetic points. However, no significant differences in the spectra were observed, so it was decided to choose the smallest time between scans that allows them to independently record each point of the kinetics with an adequate number of scans to obtain the best signal/noise ratio.

The thickness and composition of the polysiloxane hybrids were determined by field emission-scanning electron microscopy using an FE-SEM FEI Nova NanoSEM 230 (Hillsboro, OR, USA) equipped with an EDX Genesis XM2i analyzer. In order to determine both parameters, the hybrid coatings were previously deposited by dip-coating onto carbon steel substrates measuring 52 mm × 30 mm × 1 mm, withdrawn at a rate of 14 mm·min$^{-1}$ and air-dried for approximately 10 min. This procedure was executed twice, after that, the coated steels were cured at room temperature for 45 days and at 65 °C

for 4 h, conditions selected taking account the results obtained during this research. Finally, the carbon steel plates were bent in order to detach the coating from the metal surface. The coatings were observed in a cross-sectional view to determine their thickness and composition.

## 3. Results and Discussion

Initially, a characterization of the pure reagents by nuclear magnetic resonance spectroscopy was carried out. After that, the effect of the temperature on the hydrolysis and condensation mechanisms of the polysiloxane hybrids synthesized from [TMOS]/[MPTS] with a molar ratio of 1 was also studied using this technique. SEM/EDX was utilized to evaluate this effect on the thickness and composition of the formed films.

### 3.1. Pure Reagents

Figure 1 shows $^{29}$Si and $^{13}$C NMR spectra of two reagents studied: 3-methacryloxypropyltrimethoxysilane (MPTS) and tetramethyl orthosilicate (TMOS).

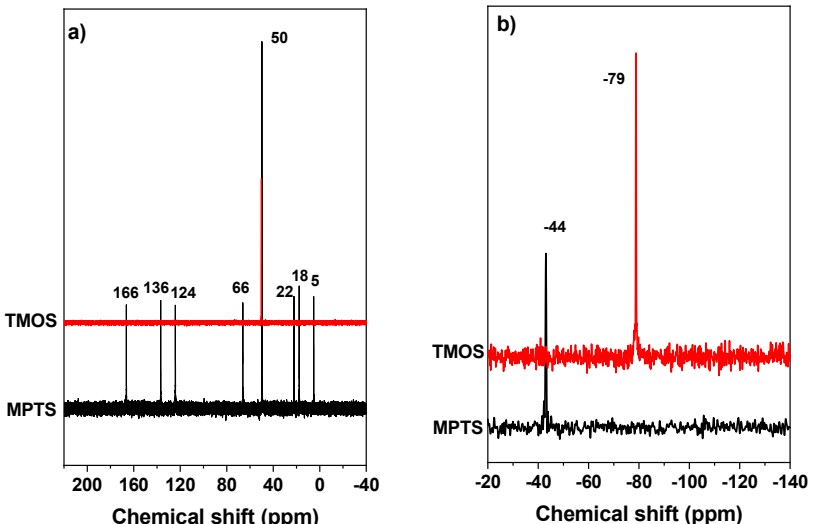

**Figure 1.** $^{13}$C (**a**) and (**b**) $^{29}$Si NMR spectra of two pure reagents.

The chemical shifts of $^{13}$C NMR spectrum for MPTS appeared at 5 ppm, associated to the first carbon linked to the silicon, at 18 ppm, carbon in the methyl terminal group, at 22 ppm, the second carbon close to the silicon, at 50 ppm, carbon in the methyl close to oxygen, at 66 ppm, the third carbon close to the silicon, at 124 and 136 ppm, vinyl group carbons ($CH_2$=C) respectively and at 166 ppm, carbonyl group carbon [29,30]. A single signal of –$CH_3$ groups at 50 ppm was observed in the spectrum of TMOS.

The chemical shift of MPTS signal detected in $^{29}$Si spectra at −44 ppm was assigned to $RCH_2$–Si(O$R$)$_3$ environments, whereas the signal of TMOS at −79 ppm was associated with Si(O$R$)$_4$ environments.

### 3.2. Network Formation of the Polysiloxane Hybrid Materials

The hydrolysis and condensation mechanisms of the polysiloxane hybrids synthesized from [TMOS]/[MPTS] mixture with and without thermal curing were studied through liquid-state $^{13}$C and $^{29}$Si NMR and solid-state $^{13}$C CP- and $^{29}$Si MAS-NMR to know how the temperature affected the network formation of these hybrid materials. The thickness and composition of the materials were determined by SEM/EDX.

Liquid-state $^{13}$C NMR spectra for the polysiloxane hybrids synthesized from [TMOS]/[MPTS] mixture at room temperature for different times are shown in Figure 2. The spectrum of $^{13}$C registered

after 1 h showed 9 signals at 8, 17, 22, 49, 57, 66, 125, 136 and 167 ± 1 ppm. The resonances at 8, 17, 22, 66, 125, 136 and 167 ppm were ascribed to MPTS, while that at 49 ppm was assigned to –CH$_3$ groups belonging to both reagents, MPTS and TMOS. In addition, the two resonances at 17 (–CH$_3$ groups) and 57 (–CH$_2$ groups) ppm were attributed to residual ethanol, used as a solvent in the sol-gel process [31].

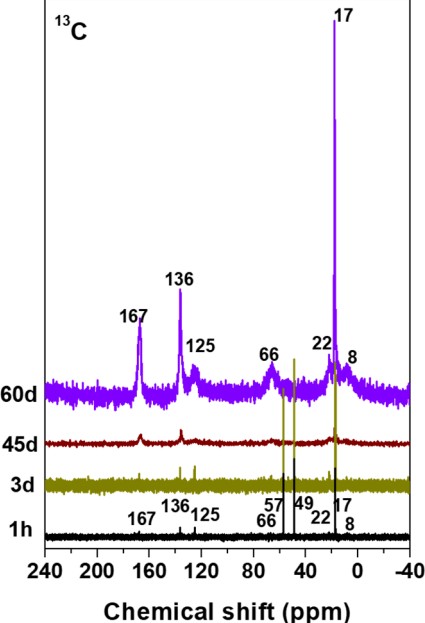

**Figure 2.** $^{13}$C NMR spectra for the polysiloxane hybrids synthesized from [TMOS]/[MPTS] mixture at room temperature for different times.

The synthesis of the hybrid materials by the sol-gel method occurs in four stages: (a) hydrolysis, (b) condensation and polymerization of monomers, (c) growth of the particles and (d) agglomeration of polymer and formation of gel [2,5]. In the first stage, a hydrolysis reaction of the silicon alkoxide groups (–O*R*) occurs. In this study, the nucleophilic attack of the negatively charged hydroxyl group to the positively charged silicon atom takes place. After that, CH$_3$OH formation occurs by transferring one proton (H$^+$) from the water to OCH$_3$ groups of Si and breaking the Si–OCH$_3$ bond. This reaction was confirmed by the increase of signal intensities in the spectrum at 49 ppm, ascribed to –CH$_3$ groups in the methanol. The signal of the carbon in the methanol appears at 49 ppm as was reported by Silverstein et al. [30].

Over time, this process was repeated successively until all the alkoxide groups are replaced, an increase of the intensity of this resonance was observed after 3 days. The rest of the spectra corresponding to the first hours were not shown in Figure 2 because they were identical to those obtained after 3 days. The hydrolysis was not completed because the spectra exhibited an appreciable amount of residual ethoxide groups (signals at 17 and 57 ppm) from the ethanol used as a solvent. After 45 and 60 days, the condensation reactions occurred between the resulting hydroxyl groups, leading the formation of a three-dimensional network, as deduced from the corresponding $^{29}$Si NMR spectra. It was reflected in $^{13}$C NMR spectra, where differences were clearly observed: signals become broader and resonances at 49 and 57 ppm disappeared. The important broadening of the peaks was an indication of a loss of mobility due to the advancement of the polycondensation.

Solid-state $^{13}$C CPMAS-NMR spectra for the polysiloxane hybrids synthesized from [TMOS]/[MPTS] mixture dried at 65 °C for different times are shown in Figure 3. The spectrum recorded after 4 h was quite similar to that obtained at room temperature after 60 days. It presented the same number and position of signals, confirming the condensation and polymerization of monomers to form chains and

particles, which grow and agglomerate to form networks that extend throughout the liquid medium resulting in a gel.

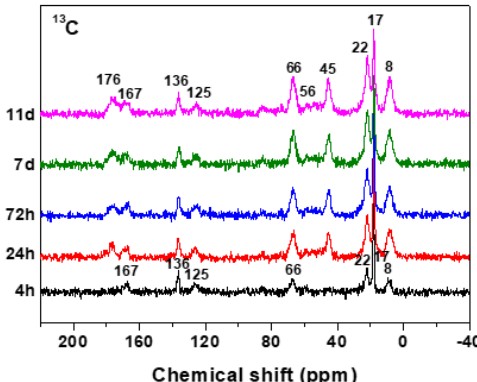

**Figure 3.** [13]C CP-MAS NMR spectra for the polysiloxane hybrids synthesized from [TMOS]/[MPTS] mixture at 65 °C for different times.

After 24 h, the spectrum showed three new resonances at 45, 56 and 176 ± 1 ppm, assigned to quaternary carbon atoms, the aliphatic –$CH_2$– groups and C=O groups bonded to aliphatic carbon groups, after the C=C opening [29]. The presence of these signals confirmed the second stage of condensation, specifically the formation of a cross-linked organic part through the vinyl groups of the methacryloxy group. Despite this, the intensity of the peaks at 167, 136 and 125 ppm remained practically constant, indicating that part of $CH_2$=C and C=O groups did not intervene in this reaction and the condensation via hydroxyl groups also took place. The spectra recorded between 3 and 11 days were of similar appearance and had the same number and position of signals.

Liquid-state [29]Si NMR spectra for the polysiloxane hybrids synthesized from [TMOS]/[MPTS] mixture at room temperature for different times are shown in Figure 4. All spectra exhibited two clearly differentiated parts, the first of them localized between −47 and −72 ppm associated with the presence of organic Si units and the second of them localized between −80 and −117 ppm attributed to the presence of inorganic Si units. [29]Si NMR spectra give information about the different Si environments formed in hybrid materials based on the number of attached carbons (T units) or silicons (Q units) and the condensation degree [30,32]. The chemical shift of silicon is determined by the chemical nature of their neighbors, namely T and Q structures. According to the nomenclature, in this study, five Q signals of different nature can be present ($Q^n$, $Si(OSi)_n(OR)_{4-n}$, where $n = 0, 1, 2, 3$ or 4, respectively and R is H or $CH_3$) and four T signals of different nature can be present ($T^{-n}$, $–CH_2Si(OSi)_n(OR)_{3-n}$, where $n = 0, 1, 2$ or 3, respectively and R is H or $CH_3$) [33].

At the beginning of the experiment, the spectrum showed five resonances at −49, −59, −82, −91 and −101 ± 1 ppm. The two former were attributed to $T^1$ ($–CH_2Si(OSi)(OR)_2$) and $T^2$ ($–CH_2Si(OSi)_2(OR)$) units, while the three latter were ascribed to $Q^1$ ($Si(OSi)(OR)_3$), $Q^2$ ($Si(OSi)_2(OR)_2$) and $Q^3$ ($Si(OSi)_3(OR)$) units [34,35]. The presence of $T^1$ and $Q^1$ species indicated that the network was poorly polymerized, there was a high number of monomers in the liquid medium. After an hour and a half, the signal at −82 ppm disappeared and the intensity of the peak at −101 ppm increased, indicating a greater polymerization degree. A new signal (−67 ppm) was exhibited in the [29]Si spectrum after 3 h, which was assigned to $T^3$ ($–CH_2Si(OSi)_3$) units. This peak indicated the formation of branched siloxane structures. With the time, the intensity of these species increased at the expense of $T^1$ species, disappearing after 3 days. The same behavior happened in the inorganic part of the spectrum, the intensity of $Q^3$ units increased at the expense of $Q^2$ units, which also disappeared after 45 days. After 45 days, the spectrum showed four resonances at −49, −67, −101 and −110 ppm. The latter was associated with $Q^4$ ($Si(OSi)_4$) units, indicating the inorganic condensation was favored. The broadening with thermal curing can be observed with all the signals due to a loss of mobility during the progress

of the polycondensation and after 60 days, the spectrum of the sample was not possible to register. The sample was so polymerized that a spectrum could not be registered without applying the magic angle spinning (MAS). The polysiloxane hybrids synthesized presented a greater disorder and a greater variation of angles and bond lengths, leading to a broadening of the peaks as well.

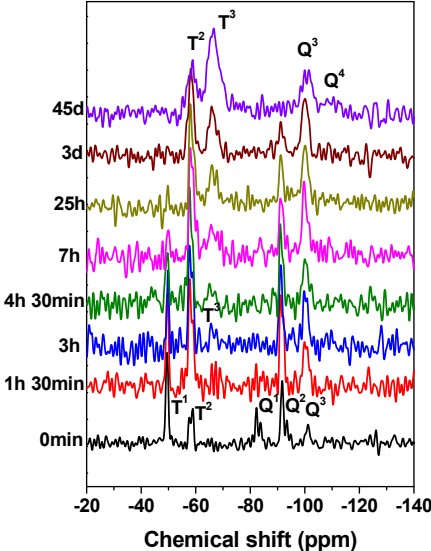

**Figure 4.** $^{29}$Si NMR spectra for the polysiloxane hybrids synthesized from [TMOS]/[MPTS] mixture at room temperature for different times.

Figure 5 shows solid-state $^{29}$Si MAS-NMR spectra for the polysiloxane hybrids synthesized from the [TMOS]/[MPTS] mixture was dried at 65 °C for different times. The spectrum registered after 4 h showed four signals. It was quite similar to that obtained at room temperature after 45 days, although the signals intensities were different. The most intense signal was that ascribed to $T^2$ unit, indicating that hybrid materials had linear structures. The rest of spectra from 24 h to 11 days presented five resonances associated with $T^1$, $T^2$, $T^3$, $Q^3$ and $Q^4$ units, where the intensity of $T^2$ and $T^3$ species tended to be equal and the intensity of $Q^4$ species increased, favoring the formation of polysiloxane hybrids with a high degree of polymerization over time.

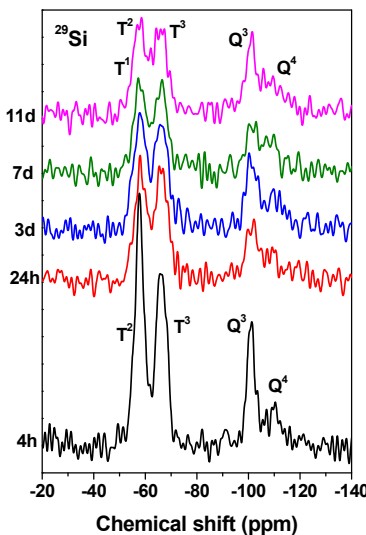

**Figure 5.** $^{29}$Si MAS NMR spectra for the polysiloxane hybrids synthesized from [TMOS]/[MPTS] mixture at 65 °C for different times.

The experimental profiles were deconvoluted with DMFIT software, and the results obtained are represented in Tables 1 and 2.

**Table 1.** Deconvolution of $^{29}$Si NMR spectra for polysiloxane hybrids synthesized from the [TMOS]/[MPTS] mixture at room temperature (RT) for different times.

| Time | Proportions (%) | | | | | | |
|---|---|---|---|---|---|---|---|
| | $T^1$ | $T^2$ | $T^3$ | $Q^1$ | $Q^2$ | $Q^3$ | $Q^4$ |
| 0 min | 30.06 | 16.46 | – | 17.36 | 26.53 | 9.58 | – |
| 1 h 30 min | 18.08 | 34.44 | – | – | 27.29 | 20.20 | – |
| 3 h | 12.04 | 33.84 | 9.71 | – | 21.06 | 23.34 | – |
| 4 h 30 min | 12.97 | 34.83 | 9.97 | – | 19.69 | 22.55 | – |
| 7 h | 4.89 | 33.82 | 17.6 | – | 17.43 | 26.26 | – |
| 25 h | – | 31.03 | 24.73 | – | 14.84 | 29.40 | – |
| 3 d | – | 32.13 | 27.81 | – | 11.91 | 28.16 | – |
| 45 d | – | 18.81 | 46.57 | – | – | 23.18 | 11.43 |

**Table 2.** Deconvolution of $^{29}$Si MAS-NMR spectra for polysiloxane hybrids synthesized from the [TMOS]/[MPTS] mixture with dried at 65 °C for different times.

| Time | Proportions (%) | | | | |
|---|---|---|---|---|---|
| | $T^1$ | $T^2$ | $T^3$ | $Q^3$ | $Q^4$ |
| 4 h | – | 39.6 | 32.36 | 16.53 | 11.51 |
| 24 h | 4.89 | 30.53 | 33.58 | 18.60 | 12.40 |
| 72 h | 8.85 | 26.56 | 30.01 | 19.66 | 14.92 |
| 7 d | 4.64 | 29.17 | 31.53 | 17.88 | 16.79 |
| 11 d | 4.84 | 26.97 | 28.96 | 23.79 | 15.44 |

At room temperature and the start of the reaction, the total amount of Q units was very similar to T units as MPTS and TMOS were introduced in a 1:1 ratio, involving an expected 50%:50% ratio. Over time, the progress of the polymerization reaction in the organic and inorganic part was seen through the relative proportions of Q and T units.

In the case of solid samples obtained by drying at 65 °C, a higher proportion of T units (70%) was obtained at the beginning of the reaction, which slowly decreased with the reaction time up to 60%.

These values were used to deduce the condensation degrees of T and Q species of these polysiloxane hybrids. The condensation degree of T and Q species was calculated according to the next expressions [34,36]:

$$TD_C\ (\%) = [D_c(T) \times T^n + D_c(Q) \times Q^n]/100 \tag{1}$$

$$D_c(T) = [(T^1 + 2T^2 + 3T^3)/3] \times 100 \tag{2}$$

$$D_c(Q) = [(Q^1 + 2Q^2 + 3Q^3 + 4Q^4)/4] \times 100 \tag{3}$$

where $T^n$ and $Q^n$ describe the relative amount of T and Q species respectively, $D_c(T)$ and $D_c(Q)$ were the condensation degree of T and Q species respectively and $TD_C$ is the total condensation degree.

Figure 6 shows the evolution of the total condensation degree, condensation degree of T species and Q species for polysiloxane hybrids synthesized with and without thermal treatment at different times. The hybrid materials synthesized without drying presented an initial total condensation degree ($TD_C$) of 46%. This value increased over time, but it did not reach a value of 75% until 3 days. At the end of the experiment, $TD_C$ was 88%. The hybrids synthesized at 65 °C showed a $TD_C$ between 80% and 83% independently of the curing time, indicating the condensation of the species was more favored from the first hours. Therefore, the total condensation degree depended on a great extent on the temperature and an initial thermal treatment obtaining a more polymerized network.

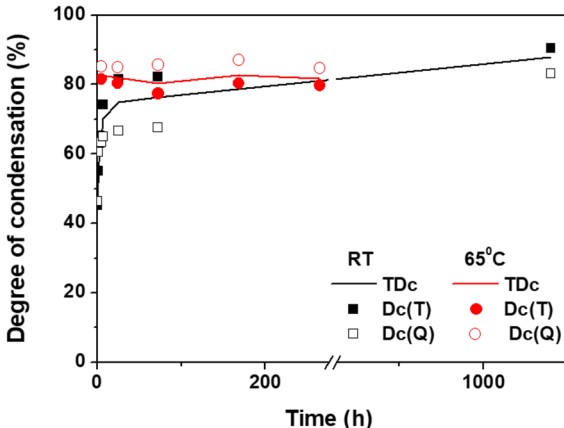

**Figure 6.** Evolution of the total condensation degree (TDc), condensation degree of T species (Dc(T)) and Q species (Dc(Q)) for polysiloxane hybrids synthesized from the [TMOS]/[MPTS] mixture at room temperature (RT) and at 65 °C for different times.

The evolution of the condensation degree of T species (Dc(T)) for polysiloxane hybrids at RT was on par with that presented for the condensation degree of Q species (Dc(Q)) during the first three hours. After that, Dc(T) was favored over time and the network of these hybrids could show two distinctive parts: organic and inorganic, being the former the predominant part. In the polysiloxane hybrids curing at 65 °C, Dc(Q) was slightly higher than that presented for Dc(T) independently of the curing time. The network of these hybrids could present a structure with organic and inorganic intermixed parts.

The dimensionality of the polysiloxane hybrids synthetized from the [TMOS]/[MPTS] mixture at room temperature (RT) and at 65 °C for different times was also determined. This parameter provides information about the final structure achieved in the hybrid materials. The total dimensionality was calculated according to the next equations [35]:

$$d_{total} \ (\%) = [d(T) \times T^n + d(Q) \times Q^n]/(T^n + Q^n) \tag{4}$$

$$d(T) = [(T^1 + 2T^2 + 3T^3)/T^n] \tag{5}$$

$$d(Q) = [(Q^1 + 2Q^2 + 3Q^3 + 4Q^4)/Q^n] \tag{6}$$

where $d(T)$ and $d(Q)$ are the total dimensionality of T and Q species and $d_{total}$ is the total dimensionality. The $d_{total}$ parameter values are between 0 and 4; when the structure is formed by isolated units has $d = 0$; when it is constituted by dimeric units $d = 1$; when linear chains are formed $d = 2$ and when the structure is bidimensional or tridimensional $d = 3$ or 4, respectively. In this study, the presence of T units implied the existence of a Si–C bond in their surroundings and therefore, the total dimensionality of the polysiloxane hybrids could never be 4.

Figure 7 shows the evolution of total dimensionality for polysiloxane hybrids synthesized with and without thermal treatment at different times. At the beginning of the synthesis (0 min), the polysiloxane hybrids synthesized at RT presented a total dimensionality ($d_{total}$) value of 1.62, indicating the structure of the material was constituted mainly by dimeric units, as observed in $^{29}$Si MAS-NMR spectrum, where low connectivity units (T$^1$ and Q$^1$) were detected. The $d_{total}$ values increased with the time, from 2.02 at an hour and a half to 2.92 at 45 days. The network was evolving throughout the experiment, initially, it was a dimeric structure, later it has constituted by linear chains and finally, it was a more condensed structure, specifically a film structure.

However, the hybrid materials synthesized at 65 °C showed a $d_{total}$ value between 2.70 and 2.78, a more complex and polymerized structure was formed and maintained over time. It is observed that the temperature also affected the total dimensionality of the polysiloxane hybrids synthesized

with TMOS and MPTS, developing a higher $d_{total}$ parameter where a thermal treatment was applied, which resulted in the development of a more complex structure.

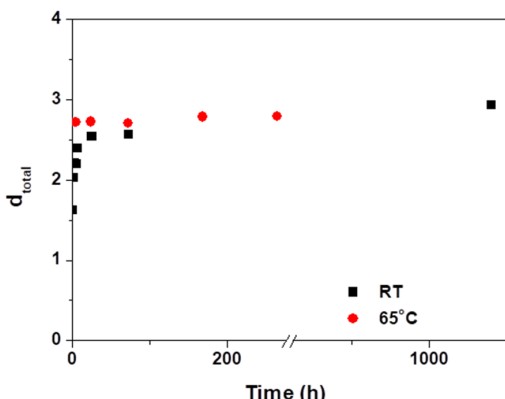

**Figure 7.** Evolution of total dimensionality for polysiloxane hybrids synthesized from the [TMOS]/[MPTS] mixture at room temperature (RT) and 65 °C for different times.

In light of these results, the network formation of the polysiloxane hybrid materials depended on the drying temperature to a large extent. In the absence of thermal treatment, the reactive hydroxyl groups generated during the hydrolysis of alkoxy groups needed long periods to form branched oligomers and polymers, and the condensation reactions did not take place until 45 days, obtaining materials with a total condensation degree lower than 75% and a $d_{total} = 2.5$. In the presence of thermal treatment, the condensation and polymerization of monomers occurred after 4 h. In addition, the second stage of condensation was favored by the presence of polymerizable vinyl groups after 24 h of curing, obtaining a material with total condensation degrees higher than 80% and a $d_{total} = 2.8$.

Taking into account the condensation degree and the total dimensionality of the hybrid coating synthesized at 65 °C for 4 h, this film could have good anticorrosive properties acting as a physical barrier against aggressive ions.

A deeper study of the hybrid coatings synthesized from the [TMOS]/[MPTS] mixture at 65 °C for 4 h and 25 °C for 45 days was carried out to know their thickness and composition. Table 3 shows these two parameters of the hybrid coatings synthesized in both curing conditions, which were determined through SEM/EDX. These certain conditions of temperature and time were chosen since synthesized hybrids seem to have a similar structure and dimensionality. The thermal treatment (65 °C) led to the formation of polysiloxane hybrids with a slightly greater thickness because of the acceleration of condensation and polymerization reactions of hydrolyzed monomers, giving a more complex and polymerized structure (Figure 8). Both SEM images showed two clearly distinguishable zones, a grey zone corresponding to the hybrid coating and a white zone associated with the carbon steel plate.

The EDX analysis showed that the main elements present in the hybrid materials were C, Si, O and Fe, the latter from the substrate. The polysiloxane hybrids synthesized at room temperature for 45 days presented a lower atomic percentage of Si, the condensation between two –OH groups or a –OH and a –*OR* group was slower and a higher number of OH groups and a lower number of Si were present in the structure of the material.

**Table 3.** Thickness and composition of the hybrid coatings synthesized from the [TMOS]/[MPTS] mixture.

| Treatment | Thickness (μm) | Composition | |
|---|---|---|---|
| | | C (%) | Si (%) |
| 25 °C for 45 d | 1.3 ± 0.2 | 48.1 ± 1.1 | 10.2 ± 0.5 |
| 65 °C for 4 h | 1.4 ± 0.5 | 51.7 ± 0.3 | 12.3 ± 1.2 |

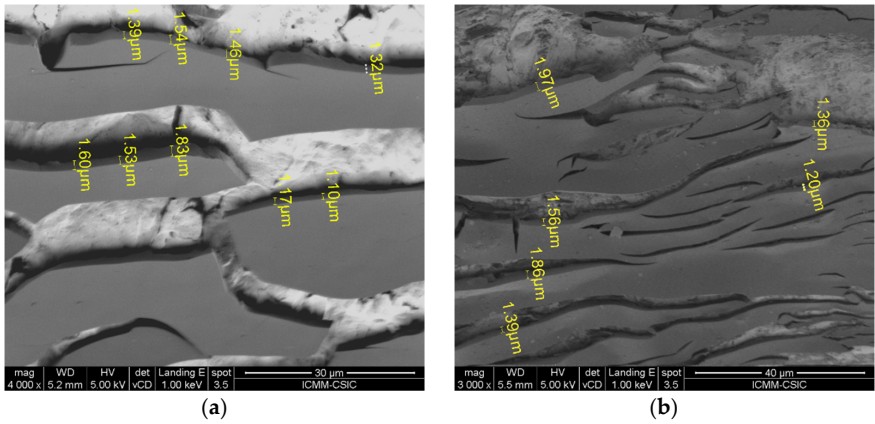

**Figure 8.** Cross-section micrograph obtained by SEM analysis for the hybrid coatings synthesized from the [TMOS]/[MPTS] mixture: (**a**) 25 °C for 45 days and (**b**) 65 °C for 4 h.

A schematic view of the chemical structure of the polysiloxane hybrids produced at room temperature for 45 days, and at 65 °C for 4 h is shown in Figure 9. The chemical structure of the coatings was performed with CrystalMarker software (version 10.5).

Comparing with the results obtained in previous works [12,17,35], the polysiloxane hybrids synthesized at 65 °C for 4 h or room temperature for 45 days presented a higher total condensation degree, low porosity and adequate thickness. All this will confer the coatings suitable characteristics as a physical barrier that will prevent the access of aggressive ions, giving them good anticorrosive properties.

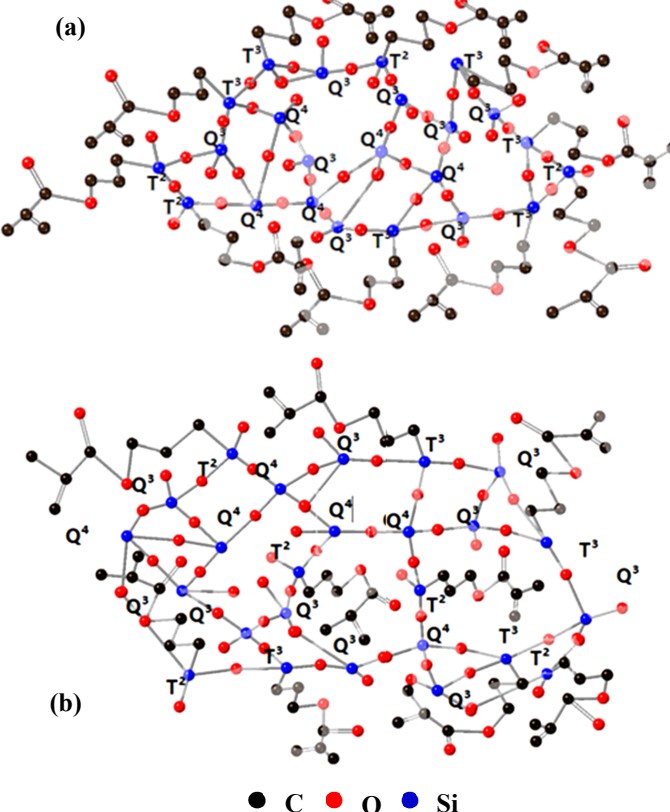

**Figure 9.** Schematic view of the chemical structure of the polysiloxane hybrids produced (**a**) at room temperature for 45 days, and (**b**) at 65 °C for 4 h. Oxygen atoms not bonded to Si atoms are bonded to H or CH$_3$ (not shown).

## 4. Conclusions

The present study shows that temperature plays an important role in the network formation of the polysiloxane hybrids via the sol-gel process. At room temperature, liquid-state NMR spectra showed that the formation of polymerized units ($T^3$ and $Q^4$) was delayed and the formation of a three-dimensional network was observed after 45 days. At 65 °C, solid-state NMR spectra showed that the condensation reactions took place in the first hours of synthesis and polymerization of organic groups via methacryloxy group occurred after 24 h.

At room temperature and the start of the reaction, the total amount of Q units ($Q^1$, $Q^2$ and $Q^3$) was very similar to T units ($T^1$ and $T^2$). Over time, the coating presented a high polymerization degree ($T^2$, $T^3$, $Q^3$ and $Q^4$ units) and it was a hybrid richer in the organic part, predominating T units.

In the samples obtained by drying at 65 °C, a higher proportion of T units (70%, $T^2$ and $T^3$ units) was obtained at the beginning of the reaction. Over time, the coating presented similar polymerization degree, prevailing $T^2$, $T^3$, $Q^3$ and $Q^4$ units, and increasing the inorganic part.

The hybrid materials synthesized at 65 °C showed a $d_{total}$ value between 2.7 and 2.8, while the $d_{total}$ values for those synthesized at room temperature increased with the time, from 2.0 at an hour and a half to 2.9 at 45 days.

The hybrid materials synthesized without drying presented an initial $TD_C$ of 46% and at the end of the experiment, it was 88%. The hybrids synthesized at 65 °C showed a $TD_C$ between 80% and 83% independently of the curing time.

The polysiloxane hybrid synthesized at 65 °C for 4 h presented a greater thickness and higher atomic percentage of Si in its composition than that synthesized at room temperature for 45 days.

These materials formed at 65 °C for 4 h show an appropriate thickness and low porosity and good adherence, which allows them to act as a physical barrier preventing the penetration of aggressive ions and decreasing the corrosion rate.

**Author Contributions:** Conceptualization, M.C. and I.S.; methodology, M.C.; software, M.C. and I.S.; validation, M.C., I.S., and J.S.; formal analysis, M.C.; investigation, M.C. and I.S.; resources, I.S.; data curation, M.C.; writing—original draft preparation, M.C.; writing—review and editing, M.C. and I.S.; visualization, M.C. and I.S.; supervision, J.S.; project administration, M.C.; funding acquisition, I.S. and J.S. All authors have read and agreed to the published version of the manuscript.

**Funding:** This research received external funding from Project MAT2016-78362-C4-2-R financed by MICINN, Spain.

**Acknowledgments:** The authors thank to Servicio de Resonancia Magnética Nuclear (ICMM-CSIC) for MAS-NMR spectra acquisition.

**Conflicts of Interest:** The authors declare no conflict of interest.

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
