# Peer review of "Polysiloxane Hybrids via Sol-Gel Process: Effect of Temperature on Network Formation"

_coatings, doi:10.3390/coatings10070677_

Round 1
Reviewer 1 Report
In this manuscript, authors prepared sol‐gel hybrids by mixing 3-methacryloxypropyltrimethoxysilane (MPTS) and tetramethyl orthosilicate (TMOS). The formation, thickness and composition of these hybrid materials were evaluated
The paper can be published in this form if the authors will made some corrections:
The novelty of the work is not clearly mentioned.
- The authors reviewed some previous works in "introduction", but what deficiencies do the authors need to continue to study? From this manuscript, what advantages does the authors' work have over others?
- For all materials, please indicated the manufacturer and concentration (e.g. 98%, TMOS, Aldrich)
- Write with space the temperatures “e.g. 65ºC – replace with 65 ºC”, pages 11, 13
- Write the text in the format imposed by the journal, page 2
- Correct the References using the Guide of the Journal
- The following references could be included to improve the quality of the manuscript:
- “Effects of the nanoassociation of hexadecyltrimethoxysilane precursors on the sol–gel process”, Journal of Sol-Gel Science and Technology, 65, 2013, 344–35
- “Influence of the hydrophobic characteristic of organo-modified precursor on the wettability of silica film”, Bulletin of Materials Science, 37, 2014, 107–115
- New Sol-gel Formulations to Increase the Barrier Effect of a Protective Coating Against the Corrosion and Wear of Galvanized Steel”, Mat. Res. 18(1), 2015, https://doi.org/10.1590/1516-1439.288914
-
“Polyacrylate/silica hybrid materials: A step towards multifunctional properties”, Journal of Dispersion Science and Technology 40(7), 2019, https://doi.org/10.1080/01932691.2018.1489276
-
“A new approach to the preparation of PDMS–SiO2 based hybrids—A structural study”, Mater. Lett., 128, 2014, 105–109.
Author Response
Authors thanks editors and reviewers for helping suggestions, which have allowed us to clarify and improve the manuscript. The comments are valuable and very helpful for revising and improving our paper, and of important guiding significance to our further researches. Revised portions are marked in the manuscript and the responses are listed as follows:
In this manuscript, authors prepared sol‐gel hybrids by mixing 3-methacryloxypropyltrimethoxysilane (MPTS) and tetramethyl orthosilicate (TMOS). The formation, thickness and composition of these hybrid materials were evaluated
The paper can be published in this form if the authors will made some corrections:
1. The novelty of the work is not clearly mentioned.
Thank you for your helpful suggestions. The introduction has been modified and the novelty of the work has been included in the manuscript.
2. The authors reviewed some previous works in "introduction", but what deficiencies do the authors need to continue to study? From this manuscript, what advantages does the authors' work have over others?
Thank you for your helpful comments. The introduction has been modified and these aspects have remarked in the manuscript.
3. For all materials, please indicated the manufacturer and concentration (e.g. 98%, TMOS, Aldrich)
Thank you for your helpful comments. This information about the reagents has been included in the text.
4. Write with space the temperatures “e.g. 65ºC – replace with 65 ºC”, pages 11, 13
Thank you for your advice. Space has been included in the temperatures.
5. Write the text in the format imposed by the journal, page 2
Thank you for your advice. The format of the text has been changed to Palatino Linotype 10.
6. Correct the References using the Guide of the Journal
Thank you for your advice. The references have been modified according to the guidelines of the journal.
The following references could be included to improve the quality of the manuscript:
“Effects of the nanoassociation of hexadecyltrimethoxysilane precursors on the sol–gel process”, Journal of Sol-Gel Science and Technology, 65, 2013, 344–35
“Influence of the hydrophobic characteristic of organo-modified precursor on the wettability of silica film”, Bulletin of Materials Science, 37, 2014, 107–115
New Sol-gel Formulations to Increase the Barrier Effect of a Protective Coating Against the Corrosion and Wear of Galvanized Steel”, Mat. Res. 18(1), 2015, https://doi.org/10.1590/1516-1439.288914
“Polyacrylate/silica hybrid materials: A step towards multifunctional properties”, Journal of Dispersion Science and Technology 40(7), 2019, https://doi.org/10.1080/01932691.2018.1489276
“A new approach to the preparation of PDMS–SiO2 based hybrids—A structural study”, Mater. Lett., 128, 2014, 105–109.
Thank you very much for your helpful comments. Some references have been utilized in the introduction to improve the quality of the manuscript.
Best regards

Reviewer 2 Report
The authors already published two articles regarding to the corrosion resistance of different coatings prepared by the sol-gel technique. From their studies the coating with the best corrosion resistance index was chosen to be further studied in this manuscript. The novelty of this study is the temperature dependent coating synthesis and the extended 13C- and 29Si- NMR studies.
With following improvements the quality of the research will be increased:
- Utilization of the selected precursor mixture have to be explained by referring to the already published articles.
- Comparative study of the corrosion resistance of the coating synthesized at room temperature and 65 ֯C need to be introduce.
- It would be good to see, how the hydrophilic/hydrophobic profile of the sample is changing with the curing temperature, I feel that somewhere around 90֯-100֯ of 2theta will be determined with the contact angle measurements, exactly at the border of hydrophilic to hydrophobic profile.
- Some extra conclusions would be welcomed.
- Line 27-28 - “an oxygen network by condensation reactions of molecular precursors” – what does oxygen network means in the quoted sentence?
- Line 68-69 – Some abnormal changes on the used font can be observed.
- Uniform style for reference writing need to be employed. For example a part of the citations used abbreviated journal description the other part full journal name.
Author Response
Authors thanks editors and reviewers for helping suggestions, which have allowed us to clarify and improve the manuscript. The comments are valuable and very helpful for revising and improving our paper, and of important guiding significance to our further researches. Revised portions are marked in the manuscript and the responses are listed as follows:
The authors already published two articles regarding to the corrosion resistance of different coatings prepared by the sol-gel technique. From their studies the coating with the best corrosion resistance index was chosen to be further studied in this manuscript. The novelty of this study is the temperature dependent coating synthesis and the extended 13C- and 29Si- NMR studies.
With following improvements the quality of the research will be increased:
1. Utilization of the selected precursor mixture have to be explained by referring to the already published articles.
Thank you very much for your helpful comments. The justification of the utilization of the selected precursor mixture has been included in the manuscript.
2. Comparative study of the corrosion resistance of the coating synthesized at room temperature and 65 ֯C need to be introduce.
Thank you very much for your helpful suggestions. The justification of the choice of both temperatures to carry out the study of the effect of the temperature in the network of the hybrids has been introduced.
3. It would be good to see, how the hydrophilic/hydrophobic profile of the sample is changing with the curing temperature, I feel that somewhere around 90֯-100֯ of 2theta will be determined with the contact angle measurements, exactly at the border of hydrophilic to hydrophobic profile.
Thank you very much for your helpful comments. The authors consider that the characterization of the coatings through NMR and SEM/EDX techniques provide enough information to know the effect of the temperature on the hydrolysis and condensation mechanisms of the polysiloxane hybrids. In the future, a study of the hydrophilic/hydrophobic profile of the coatings will also be considered.
4. Some extra conclusions would be welcomed.
Thank you very much for your helpful comments. Some extra conclusions have been included.
5. Line 27-28 - “an oxygen network by condensation reactions of molecular precursors” – what does oxygen network means in the quoted sentence?
Thank you very much for your helpful suggestions. The sentence has been modified.
6. Line 68-69 – Some abnormal changes on the used font can be observed.
Thank you for your advice. The format of the text has been changed to Palatino Linotype 10.
7. Uniform style for reference writing need to be employed. For example a part of the citations used abbreviated journal description the other part full journal name.
Thank you for your advice. All the journals have been abbreviated rightly.
Best regards

Reviewer 3 Report
The article is devoted to the study of sol-gel coatings via NMR spectroscopy. The work is scientifically saturated but flawed in reasoning and motivation of the research. The studied system contained thermal initiator of polymerization (BPO), it is unsurprising that the thermally treated system will show better performance. Technically the article resembles the other work of the authors (DOI: 10.1016/j.porgcoat.2014.01.019, uncited). In order to increase the quality of the work the following points should be addressed:
1. The Introduction section should be adjusted to represent the content of the research.
2. It is unclear how the authors determined the thickness of the coatings via SEM and EDX. The quality of Figure 8 is to low to make any conclusions.
3. Lines 355-360 are not supported by the research results: it is impossible to make a conclusion that 65 oC is the ideal conditions as other temperatures were not tested. It is unclear why such systems "probably show an appropriate thickness and low porosity and good adherence".
4. There are some typos in the text
Author Response
Authors thanks editors and reviewers for helping suggestions, which have allowed us to clarify and improve the manuscript. The comments are valuable and very helpful for revising and improving our paper, and of important guiding significance to our further researches. Revised portions are marked in the manuscript and the responses are listed as follows:
The article is devoted to the study of sol-gel coatings via NMR spectroscopy. The work is scientifically saturated but flawed in reasoning and motivation of the research. The studied system contained thermal initiator of polymerization (BPO), it is unsurprising that the thermally treated system will show better performance. Technically the article resembles the other work of the authors (DOI: 10.1016/j.porgcoat.2014.01.019, uncited). In order to increase the quality of the work the following points should be addressed:
1. The Introduction section should be adjusted to represent the content of the research.
Thank you very much for your helpful suggestions. The introduction has been modified and the content of the research has been included.
2. It is unclear how the authors determined the thickness of the coatings via SEM and EDX. The quality of Figure 8 is to low to make any conclusions.
Thank you very much for your helpful comments. The determination of the thickness of the coatings has been explained in the text and a description of the micrographs (coating and steel) has been included as well. A great number of images was used to determine the thickness of the coatings and very small differences were found between the measurements.
3. Lines 355-360 are not supported by the research results: it is impossible to make a conclusion that 65 oC is the ideal conditions as other temperatures were not tested. It is unclear why such systems "probably show an appropriate thickness and low porosity and good adherence".
Thank you very much for your helpful comments. This paragraph has been removed in the manuscript.
4. There are some typos in the text
Thank you for your advice. The authors have used Grammarly software to correct the typos in the manuscript.
Best regards

Reviewer 4 Report
Dear Authors,
I think that the manuscript (MS) can be published after some corrections.
In the following, some points to improve and correct the MS are given. Please use these suggestions.
Abstract
1. Lines 11-13: Please rewrite these lines. Are you sure that this coatings have excellent corrosion properties?
2. Lines 20-21: “dtoyal” is dtotal , but an abbreviation in the abstract is not advisable. Moreover here it is not declared.
Keywords
3. Temperature is too generic as keyword.
Dimensionality is not a keyword.
Introduction
4. Lines 68-69: “In light of these results, this research was proposed to study the effect of the temperature on the network formation of polysiloxane hybrids as a possible physical barrier to prevent corrosion.”
The research is not proposed, but aimed.
About the effect of the temperature:
- Why did you study only one temperature?
- Why did you choose 65 °C?
Put the answers also in the text.
5. Lines 95-96: “on the other hand” must be changed to “in the second one”.
Experiments
6. Please substitute “ºC” with “°C” in the whole MS.
Results and discussion
7. Lines 127-130: rewrite in a more understandable way. You declared that the effect of temperature is your primary aim. Here you start to study a typical synthesis steps and not your primary aim. … performed not took place …
8. Please choose univocally the way to cite the figure: Fig. or Figure.
9. Lines 208-209: “remained practically constant, no decreased drastically” please rewrite in a more scientific way.
10. Lines 210-211: Change “The spectra recorded between 3 and 11 days displayed similar appearance, same number and position of signals.” with “The spectra recorded between 3 and 11 days are similar.”
11. Line 250: .. tend to BE equal …
12. In tables 1 and 2, there are “a” after “Proportions” that is not described.
13. Figure 9: you must declare the program used to obtain the images. You must improve the quality of the present figures.
Conclusions
14. Please write and underline here the positive effect of thermal treatment onto the coatings for a final applications. You must put (or shortly repeat) lines 355-360 (now in the previous section) in the conclusion.
References
15. The references could be improved with some more recent citations.
Best regards
Author Response
Authors thanks editors and reviewers for helping suggestions, which have allowed us to clarify and improve the manuscript. The comments are valuable and very helpful for revising and improving our paper, and of important guiding significance to our further researches. Revised portions are marked in the manuscript and the responses are listed as follows:
Dear Authors,
I think that the manuscript (MS) can be published after some corrections.
In the following, some points to improve and correct the MS are given. Please use these suggestions.
Abstract
1. Lines 11-13: Please rewrite these lines. Are you sure that this coatings have excellent corrosion properties?
Thank you very much for your helpful comments. These lines have been rewritten. These coatings have corrosion properties as was demonstrated in several studies carried out by E. Bakhshandeh et al. Prog. Org. Coat. 2014; X.F. Yang et al. Surf. Coat. Technol. 2001; N.N. Voevodin et al. Surf. Coat. Technol. 2006; S.R. Kunst et al. Mater Res. 2015 among others.
2. Lines 20-21: “dtoyal” is dtotal , but an abbreviation in the abstract is not advisable. Moreover here it is not declared.
Thank you very much for your helpful comments. The abbreviation has been changed and the meaning of this abbreviation has been included in the abstract.
Keywords
3. Temperature is too generic as keyword.
Dimensionality is not a keyword.
Thank you very much for your helpful suggestions. Both keywords have been modified and specific terms have included in the manuscript.
Introduction
4. Lines 68-69: “In light of these results, this research was proposed to study the effect of the temperature on the network formation of polysiloxane hybrids as a possible physical barrier to prevent corrosion.”
The research is not proposed, but aimed.
About the effect of the temperature:
Why did you study only one temperature?
Why did you choose 65 °C?
Put the answers also in the text.
Thank you very much for your helpful comments. This paragraph has been modified, the research has been proposed in the manuscript. Besides, the choice of temperatures (65 ºC and room temperature) has been justified according to previous studies.
5. Lines 95-96: “on the other hand” must be changed to “in the second one”.
Thank you for your advice. This structure has been removed in the manuscript.
Experiments
6. Please substitute “ºC” with “°C” in the whole MS.
Thank you for your advice. It has been substituted in the manuscript.
Results and discussion
7. Lines 127-130: rewrite in a more understandable way. You declared that the effect of temperature is your primary aim. Here you start to study a typical synthesis steps and not your primary aim. … performed not took place …
Thank you very much for your helpful comments. This paragraph has been rewritten identifying the steps following in the study: characterization of pure reagents and the effect of temperature on the network formation of the coatings regarding the polymerization degree and total dimensionality, thickness and composition.
8. Please choose univocally the way to cite the figure: Fig. or Figure.
Thank you for your advice. The way to cite the figures has been changed.
9. Lines 208-209: “remained practically constant, no decreased drastically” please rewrite in a more scientific way.
Thank you very much for your helpful comments. The sentence has been modified.
10. Lines 210-211: Change “The spectra recorded between 3 and 11 days displayed similar appearance, same number and position of signals.” with “The spectra recorded between 3 and 11 days are similar.”
Thank you for your advice. The sentence has been modified.
11. Line 250: .. tend to BE equal …
Thank you for your advice. BE has been included in the manuscript.
12. In tables 1 and 2, there are “a” after “Proportions” that is not described.
Thank you for your advice. “a” has been deleted in the tables.
13. Figure 9: you must declare the program used to obtain the images. You must improve the quality of the present figures.
Thank you very much for your helpful suggestions. The software used to obtain the chemical structure of the coatings has been included in the manuscript and the quality of the figure has been improved.
Conclusions
14. Please write and underline here the positive effect of thermal treatment onto the coatings for a final applications. You must put (or shortly repeat) lines 355-360 (now in the previous section) in the conclusion.
Thank you very much for your helpful suggestions. A new conclusion has been added regarding the positive effect of thermal treatment on the coating in its final applications.
References
15. The references could be improved with some more recent citations.
Thank you very much for your helpful comments. More recent references have been included in the manuscript.
Best regards

Round 2
Reviewer 1 Report
The authors have improved the manuscript and it can be published in this form.
Author Response
Thank you very much for your help.
Reviewer 2 Report
1. Measurements regarding to anticorrosive properties of the synthesized coatings need to be introduced. Also a discussion about the cross linking level and anticorrosive properties would be welcomed.
2. Line 12-13 - „In the last decades, the deposition of these coatings on different substrates, mostly metals, has demonstrated excellent corrosion properties” – may be: excellent anticorrosion properties or excellent corrosion protection or excellent corrosion resistance.
3. Line 36-38 – „Some advantages such as mild processing conditions, high purity, homogeneity of products and the possibility of modifying the process conditions are presented by the sol‐gel process in the synthesis of organic‐inorganic materials [9].” Need to be rephrased: Synthesis of the organic-inorganic materials by the sol-gel process offers some advantages such as….
4. Line 38-41, Line 88-93, line 95-97 - The phrases are ambiguous – please reformulate or rephrase
5. Line 153 -154 – should be corrected like this: … to detach the coating from the metal surface. The coatings were observed in a cross sectional view to determine their thickness and composition.
6. The conclusions still not covers all the presented results.
7. All the newly introduced sentences should be checked according their English coherence.
Author Response
Authors thanks editors and reviewers for helping suggestions, which have allowed us to clarify and improve the manuscript. Revised portions are marked in the manuscript and the responses are listed as follows:
1. Measurements regarding to anticorrosive properties of the synthesized coatings need to be introduced. Also a discussion about the cross linking level and anticorrosive properties would be welcomed.
Thank you very much for your helpful comment. The main objective of the manuscript was to study the effect of temperature on the hydrolysis and condensation mechanisms of the polysiloxane hybrids synthesized from [TMOS]/[MPTS] with a molar ratio of 1 by NMR and SEM/EDX, and the anticorrosive properties were not consider in this manuscript. However, the authors have incorporated a paragraph comparing the results obtained in their previous works to those obtained in this manuscript and an interpretation of the anticorrosive properties has been done.
2. Line 12-13 - „In the last decades, the deposition of these coatings on different substrates, mostly metals, has demonstrated excellent corrosion properties” – may be: excellent anticorrosion properties or excellent corrosion protection or excellent corrosion resistance.
Thank you very much for your advice. The sentence has been modified in the manuscript.
3. Line 36-38 – „Some advantages such as mild processing conditions, high purity, homogeneity of products and the possibility of modifying the process conditions are presented by the sol‐gel process in the synthesis of organic‐inorganic materials [9].” Need to be rephrased: Synthesis of the organic-inorganic materials by the sol-gel process offers some advantages such as….
Thank you very much for your helpful comments. The paragraph has been rewritten.
4. Line 38-41, Line 88-93, line 95-97 - The phrases are ambiguous – please reformulate or rephrase
Thank you very much for your helpful comments. All of these paragraphs have been rewritten throughout the manuscript.
5. Line 153 -154 – should be corrected like this: … to detach the coating from the metal surface. The coatings were observed in a cross sectional view to determine their thickness and composition.
Thank you very much for your advice. The sentence has been modified in the manuscript.
6. The conclusions still not covers all the presented results.
Thank you very much for your helpful suggestions. Two new conclusions have been incorporated.
7. All the newly introduced sentences should be checked according their English coherence
Thank you very much for your advice. The new sentences have been checked according to their English coherence.
Reviewer 3 Report
The article may be published in the present form.
Author Response
Thank you very much for your help.
Round 3
Reviewer 2 Report
With the improvements made by the authors, the article can be published in the present form.